# Evaluation of Tacrolimus’ Adverse Effects on Zebrafish in Larval and Adult Stages by Using Multiple Physiological and Behavioral Endpoints

**DOI:** 10.3390/biology13020112

**Published:** 2024-02-10

**Authors:** Wen-Wei Feng, Hsiu-Chao Chen, Gilbert Audira, Michael Edbert Suryanto, Ferry Saputra, Kevin Adi Kurnia, Ross D. Vasquez, Franelyne P. Casuga, Yu-Heng Lai, Chung-Der Hsiao, Chih-Hsin Hung

**Affiliations:** 1Institute of Biotechnology and Chemical Engineering, I-Shou University, Kaohsiung 84001, Taiwan; wwfeng@ntu.edu.tw (W.-W.F.); hsiuchsochen@gmail.com (H.-C.C.); 2Department of Dermatology, E-Da Cancer Hospital, Kaohsiung 82445, Taiwan; 3Dr. Feng’s Dermatology Clinic, Kaohsiung 82445, Taiwan; 4Department of Chemistry, Chung Yuan Christian University, Chung-Li, Taoyuan 320314, Taiwan; gilbertaudira@yahoo.com (G.A.); michael.edbert93@gmail.com (M.E.S.); ferrysaputratj@gmail.com (F.S.); kevinadik-adi@hotmail.com (K.A.K.); 5Department of Bioscience Technology, Chung Yuan Christian University, Chung-Li, Taoyuan 320314, Taiwan; 6Research Center for Natural and Applied Sciences, Department of Pharmacy, University of Santo Tomas, Manila 1008, Philippines; rdvasquez@ust.edu.ph (R.D.V.); fpcasuga@ust.edu.ph (F.P.C.); 7The Graduate School, Faculty of Pharmacy, University of Santo Tomas, Manila 1008, Philippines; 8Department of Chemistry, Chinese Culture University, Taipei 11114, Taiwan; lyh21@ulive.pccu.edu.tw; 9Research Center for Aquatic Toxicology and Pharmacology, Chung Yuan Christian University, Chung-Li, Taoyuan 320314, Taiwan

**Keywords:** zebrafish, tacrolimus, toxicity, behaviors, memory

## Abstract

**Simple Summary:**

Despite the clinical usefulness of tacrolimus in preventing organ transplant rejection, numerous studies have demonstrated its adverse effects in humans and animals. However, to date, no study has comprehensively addressed its toxicity toward zebrafish. Using various tests to assess the cardiovascular performance, various behaviors, and cognitive performance of the fish, we found that tacrolimus in relatively low concentrations altered the behaviors, social cohesion, and short-term memory of zebrafish in the larval and adult stages after acute and chronic exposures, respectively, with some alterations occurring in a dose-dependent manner. We hypothesize that tacrolimus may possess anxiotropic properties, which make it capable of modifying anxiety in the fish by affecting their nervous system. Nevertheless, considering the potential toxicity of tacrolimus, its usage, including in clinical applications, should be taken into account and monitored.

**Abstract:**

Tacrolimus (FK506) is a common immunosuppressant that is used in organ transplantation. However, despite its importance in medical applications, it is prone to adverse side effects. While some studies have demonstrated its toxicities to humans and various animal models, very few studies have addressed this issue in aquatic organisms, especially zebrafish. Here, we assessed the adverse effects of acute and chronic exposure to tacrolimus in relatively low doses in zebrafish in both larval and adult stages, respectively. Based on the results, although tacrolimus did not cause any cardiotoxicity and respiratory toxicity toward zebrafish larvae, it affected their locomotor activity performance in light–dark locomotion tests. Meanwhile, tacrolimus was also found to slightly affect the behavior performance, shoaling formation, circadian rhythm locomotor activity, and color preference of adult zebrafish in a dose-dependent manner. In addition, alterations in the cognitive performance of the fish were also displayed by the treated fish, indicated by a loss of short-term memory. To help elucidate the toxicity mechanism of tacrolimus, molecular docking was conducted to calculate the strength of the binding interaction between tacrolimus to human FKBP12. The results showed a relatively normal binding affinity, indicating that this interaction might only partly contribute to the observed alterations. Nevertheless, the current research could help clinicians and researchers to further understand the toxicology of tacrolimus, especially to zebrafish, thus highlighting the importance of considering the toxicity of tacrolimus prior to its usage.

## 1. Introduction

Tacrolimus is also known as FK506 due to its high affinity with the FK506-binding protein (FKBP), which is an immunophilin, making it a very potent immunosuppressive agent [1]. The mechanism of action of tacrolimus as an immunosuppressant focuses on the suppression of the T cell transcription of the Nuclear factor of activated T-cells (NF-AT) through binding with immunophilin FK-binding protein (FKBP). Tacrolimus is proven to block the action of the enzyme calcineurin phosphatase, which irreversibly inhibits the development and function of the immune system, specifically T cells, cytokines, and interleukins; it has also been approved for use as a prophylaxis against organ transplant rejection and is also prescribed for several auto-immune diseases. Furthermore, it is also recommended as a prophylaxis for graft-versus-host disease (GVHD) after hematopoietic stem cell transplant (HSCT) [2]. Unfortunately, although tacrolimus has been the drug of choice for reducing the risk of organ transplant rejection, there are several adverse effects associated with its usage either acutely or chronically, including nephrotoxicity, neurotoxicity, metabolic disorders, and gastrointestinal effects, among others [3]. Furthermore, long-term exposure to tacrolimus may lead to significant aggressiveness or marked anxious behavior, as well as neuropsychological effects like depression, significant weight loss, and insomnia [4]. Although the pharmacokinetics of tacrolimus are well studied and established, there is still a need to focus on its observable pharmacodynamic effects, especially behavioral or neuropsychological effects [5].

To date, the scientific community has focused on several significant topics addressing major health problems and highlighting the importance of the use of animal models in novel discoveries regarding the treatment of diseases. With the use of rodents, monkeys, pigs, and even marine or aquatic models (e.g., zebrafish), several health problems such as cancer, heart disease, and brain disorders (neurological as well as behavioral) have been explored with the use of newly developed approaches [6]. In animals, some studies have addressed the toxicities of tacrolimus in various animal models, as mentioned in Table 1. However, in aquatic animals, especially fish, only a few studies have demonstrated its effect on the fish. For example, Khoei et al. found that the effects of tacrolimus in Siamese fighting fish caused a significant reduction in their aggression level, which was indicated by fewer opercular displays in a mirror test after tacrolimus treatment [7]. Meanwhile, a study in zebrafish larvae revealed that tacrolimus significantly induced hyperactivity and hyperexcitability in the presence of sound and that it can suppress normal behaviors. Taken together, these results strongly suggest that fish are an effective model for initial safety and efficacy tests of other dual-specificity tyrosine-phosphorylation-regulated kinase (DYRK) inhibitors that target calcineurin signaling restoration in neural development; they thus provide promising innovative models to better understand the effects of tacrolimus in vertebrates, even including humans to a greater extent [8].

In biomedical research, zebrafish offer numerous advantages over those other animal models used to represent human diseases, specifically in large-scale toxicity screening, the pharmacologic and therapeutic screening of compounds, as well as genetic mutant applications; this is because they provide reliable results comparable to humans due to the high similarity between their genetic makeup and humans’ [9]. Furthermore, the central nervous system of adult zebrafish is similar to that of other higher forms of vertebrates and is well illustrated in different stages of animal life [10]. In addition, the various structures and subdivisions of the human brain are also present in zebrafish [11], confirming a strong correlation between human and zebrafish brain functions in relation to behavioral studies [12]. Thus, the use of zebrafish as a disease animal model in developmental disorders, behavioral changes, and mental disorders is significant in providing promising innovation in the diagnosis and treatment of diseases, as well as in discovering the effects of the chronic use of drugs [6]. However, despite these advantages, to date, there no study has comprehensively assessed the toxicities of tacrolimus in zebrafish in both larval and adult stages.

Since the observable effects of tacrolimus in zebrafish have not yet been comprehensively evaluated, the current study aimed to evaluate the adverse effects of tacrolimus in relatively lower concentrations than previous studies in both larval and adult zebrafish [7,13]. We hypothesized that even in a low concentration, tacrolimus could cause adverse effects in zebrafish in both stages. The detailed experiment protocols regarding the tacrolimus exposure and toxicity test schedule can be found in Figure 1.

**Table 1 biology-13-00112-t001:** Summary of potential adverse effects of tacrolimus exposure in experimental animals and humans.

Models	Dose	Route of Administration	Adverse Effects	References
Human	Not described	Intravenous	The most common side effects were headache, nausea, vomiting, and flushing.	[14]
	0.15 mg/kg	Intravenous, followed by oral dosing	Deterioration in renal function during a 12-month follow-up was observed. Severe nephrotoxic reactions occurred early after transplantation, often during intravenous administration of the drug in some patients.	[15]
	0.075 mg/kg	Intravenous	High incidence of moderate or severe CNS toxicity in the treated patients with longer duration of symptoms compared to cyclosporine A.	[16]
Baboon and dog	0.1–2 mg/kg/d	Intramuscular (baboon) Oral (dog)	Anorexia and lethargy, lethal emaciation, and hyperglycemia.	[17]
Cynomolgus monkey	1–10 mg/kg/d	Oral or intramuscular	Intramuscular: panlobular micro- and macrovesicular steatosis of hepatocytes, cardiomyopathy, hyalination of pancreatic islets, acute tubular necrosis. Oral: significant weight reduction, centrilobular micro- and macrovesicular steatosis of hepatocytes.	[18]
Dog	0.08–1 mg/kg/d	Oral or intramuscular	Vascular changes in the heart and irregular nuclear-shaped cells in the proximal tubules of the kidney.	[19]
Rabbit	1 mg/kg/d	Intramuscular injections	Ten of the 12 animals in the original group died or required euthanasia. At necropsy, renal failure, cardiac abnormalities, and pulmonary edema were found.	[20]
	10–1000 µg	Injection into the mídvitreous cavity of the eyes	Transient vitreous opacities were observed in the groups that received 100 and 250 µg, while 500 and 1000 µg of tacrolimus proved to be toxic to the retina.	[21]
Rat	0.5–4 mg/kg/d	Oral	Thymic medullary atrophy that appeared to be proportional to dosage; hyperglycemia.	[22]
	1 mg/kg/d	Intramuscular injections	Thymic medullary atrophy.	[23]
*Betta splenden*s	0.05 and 0.1 µg/mL	Waterborne exposure	Tacrolimus caused alterations in aggression and immunity indexes.	[7]
*Danio rerio*	0.1 µg/mL	Waterborne exposure	Coordinated and allometric fin overgrowth.	[13]

## 2. Material and Methods

### 2.1. Zebrafish Husbandry

For the larval experiments, the larvae were obtained from the breeding of AB zebrafish. Prior to the breeding process, the adult zebrafish were reared in a continuous aerated water system filtered with UV light at 26 ± 1 °C, with a 14/10 h light/dark cycle. The pH and conductivity of the water were maintained at 7.0–7.5 and between 300 and 1500 μS, respectively. The fish were fed two times a day with lab-grown brine shrimp and once daily with commercial dry food (Taiwan Hung Kuo Industrial Co., Ltd., Taipei, Taiwan). The applied animal husbandry protocol was based on a prior publication [24]. For the breeding process, two male and one female fish were used. After the mating process, the fertilized eggs were collected in a petri dish filled with sterile water and methylene blue (0.0001%), which acted as a fungicide, and kept in an incubator at 28 ± 1 °C. Afterwards, only the healthy embryos were kept and used in the following experiments.

Next, for the experiments in adult zebrafish, fish of similar size and age were obtained through an aquarium vendor from Zgenebio Inc. in Taipei, Taiwan (https://www.zgenebio.com.tw/english.html (accessed on 29 January 2024)) and acclimated for at least 1 month in the same rearing conditions mentioned above. Later, the healthy zebrafish with a normal morphology were used in the following experiments. The experiment was conducted according to the guidelines for the care and use of Laboratory Animals by Chung Yuan Christian University (CYCU) and approved by the Animal Ethics Committee of CYCU.

### 2.2. Chemical Delivery

Tacrolimus in powder form was purchased from Aladdin (Shanghai, China) with ≥98% purity. Later, in preparing a 1% stock solution, it was dissolved in DMSO; meanwhile, for the working solution in 100 ppm, the stock solution was diluted in ddH_2_O. Afterward, all of the larval and adult zebrafish were systemically exposed to tacrolimus in 0 (control), 1, and 100 ppb. These concentrations were chosen to evaluate whether tacrolimus in these relatively low concentrations had already caused toxicities to the fish, since these doses were lower than the concentrations applied in previous studies [7,13]. In addition, these low doses were also closer to the environmentally relevant concentrations described in a prior study [25]. For the larval study, DMSO in 0.01% was also added to the control group, considering the higher sensitivity of larvae to pollutants than adults [26]. This concentration of DMSO was based on previous studies that demonstrated the toxicity of this common solvent in zebrafish larval locomotion, and the cardiovascular performance was observed in a concentration of ≥0.55% and ≥1.0%, respectively [27,28]. Details regarding the exposure and toxicity evaluation of tacrolimus in the present study can be found in Figure 1.

### 2.3. Cardiac Performance Assays in Zebrafish Larvae

Zebrafish larvae were exposed to tacrolimus at 2 dpf for 1 day. Afterward, the tested larvae were mounted in 3% methylcellulose to minimize their movement prior to the video recording. For cardiac rhythm and physiology assessments, a high-resolution 4K charged coupling device (CCD; XP4K8MA, ToupTek, Hangzhou, China) mounted on an upright microscope (EX20, SOPTOP, Taipei, Taiwan) was used to record the ventricle chamber of the larvae for one minute with a video resolution of 1920 × 1080 at 30 frames per second (fps). From the videos, the interval of each ventricle beat of the larvae was calculated by measuring the dynamic pixel intensity using Time Series Analyzer V3 Plug-in on ImageJ Software v1.52 (https://imagej.nih.gov/ij/plugins/time-series.html (accessed on 11 December 2023)). Meanwhile, the sd1 and sd2 of the ventricle chamber beat intervals were generated from a point care plot using OriginPro 2019 Software v9.6.5.169 (Originlab Corporation, Northampton, MA, USA); these were used to calculate the heartbeat variability. Later, to evaluate their cardiac physiology, the lengths of the short and long axes of the heart chamber during the diastolic and systolic phases were measured to obtain the end-diastolic and systolic volumes by assuming that the heart chamber had a spheroid shape [29]. The cardiac endpoints were calculated later by using the formulas described in a prior study [30]. Finally, a high-speed digital CCD camera (AZ Instrument, Taichung City, Taiwan) combined with a Hoffman modulation contrast objective lens with 40× mounted on an inverted microscope (ICX41, Sunny Optical Technology, YuYao, China) was used to record the dorsal aorta area of the larvae during the vascular performance analysis for 10 s at 200 fps. Afterward, the “Trackmate” plugin in ImageJ was utilized to calculate the blood flow velocity; then, the diameter of the dorsal aorta was calculated using ImageJ. To obtain the blood flow profile, a file consisting of two rows of data (time and blood flow speed per frame for each detected blood cell) of one of the most represented larvae was smoothed using the smooth function from Origin Pro 2019 and output as a graph later by GraphPad Prism (GraphPad Software version 8 Inc., La Jolla, CA, USA). All the tests were performed in three replications and the applied protocol was based on a previous study [30].

### 2.4. Respiratory Rate Assay in Zebrafish Larvae

Prior to the test, 3 dpf larvae were incubated with tacrolimus. After one day of incubation, the larvae were transferred to a 24-well plate equipped with a Sensor Dish Reader (SDR) by Loligo Systems (Loligo Systems, Viborg, Denmark); this had one empty well, which served as a blank. After 10 min, the oxygen level in each well was measured every minute for 75 min using MicroResp™ software version 1.0.4 (Loligo Systems, Viborg, Denmark). This test was carried out in three replications and the current protocol was based on prior publications [31,32].

### 2.5. Light-Dark Locomotion and Vibrational Startle Response Assays in Zebrafish Larvae

One day after being exposed to tacrolimus, the 4 dpf larvae were transferred to a 48-well transparent plastic plate, one in each well, for the following light–dark locomotion test. The plate was later put into Zebrabox (Viewpoint, Civrieux, France, https://www.viewpoint.fr/product/zebrafish/fish-behavior-monitoring/zebrabox (accessed on 7 February 2024)) for 30 min to habituate with the novel environment and then later, during the test, their movements were recorded and tracked by using the ViewPoint system (version 1.3), a video-tracking software (Viewpoint, Civrieux, France). The test consisted of four 20 min cycles of alternating light and dark periods with 10 min for each period; meanwhile, the tracking process was set to one-minute intervals. The applied protocol was based on previous studies [33,34]. On the following day, the exposed larvae were subjected to the vibrational startle response assay (VSRA). This assay utilized the same instruments as the ones utilized in the light–dark locomotion test. However, in this test, tapping stimulus was used to observe the behavior responses of the larvae. After 30 min of habituation, the test was started. The first 5 s were without any stimulus, followed by 20 s of tapping stimulus at the highest intensity with a 1 s interstimulus interval. Finally, the ViewPoint system tracked the movements of the larvae and calculated the distance they traveled every second. The current protocol of this test was based on a prior publication and performed in three replications [35].

### 2.6. Multiple Behaviors Test in Adult Zebrafish

#### 2.6.1. Novel Tank Test

For the novel tank, the adaptability of the fish in a new environment was observed by introducing the zebrafish individually to a novel environment, which is a trapezoid-shaped tank (28 × 5 × 15 cm) filled with ~1.25 L of water. The movement was recorded immediately for one minute at 5 min intervals until 30 min had passed. The behaviors of the tested fish and their response to the novel environment were evaluated by calculating their locomotion, movement orientation, and exploratory behavior-related endpoints.

#### 2.6.2. Shoaling Test

Next, a shoaling test was performed to observe shoaling behavior; this is one of the most common social behaviors found in a group of fish, reducing anxiety and providing protection from predators. An identical tank configuration with a novel tank test was used in this test, and zebrafish in groups of three were introduced into the new tank. Contrary to the novel tank test, fish were acclimated for ~1 min and their behaviors were recorded for 5 min. In evaluating the shoaling formation of the tested fish, some behavior endpoints were related to the shoal size and the distance between each tested fish were calculated.

#### 2.6.3. Aggressiveness Test

On the following day, a mirror biting test was conducted to determine the aggressiveness of the zebrafish by using the same tank as that used in the novel tank test, with the addition of a mirror on one side of the tank. Fish were also acclimated for ~1 min, followed by recording for 5 min. Aggressive behavior was evaluated by observing their mirror-biting behaviors and accelerated swimming tendencies.

#### 2.6.4. Conspecific Social Interaction Test

The aggressiveness test was followed by a conspecific social interaction test, which assessed the interaction between two individual zebrafish. This test was conducted in the same tank as the novel tank test, with a slight modification, which was the placement of a transparent glass separator in the middle of the tank to separate the fish. Similar to previous methods, the tested fish were acclimated for ~5 min, followed by the introduction of their conspecific on the other side; subsequently, the behavior responses of the tested fish toward their conspecific were recorded for 5 min.

#### 2.6.5. Fear Response Test

Finally, a fear response test was carried out on the following day to observe the behavior of the zebrafish in the presence of a predator. This test was simulated in the identical tank configuration as the conspecific social test; only one side was occupied by a convict cichlid (*Amatitlania nigrofasciata*) as the fear stimulus for the tested fish. The zebrafish were acclimated for ~5 min, then the convict cichlid was introduced to the other side of the separator; the fear response behaviors of the tested zebrafish, which are usually displayed by a longer distance between the tested fish and the convict cichlid, were then recorded for 5 min.

All of the fish behaviors in each behavior test were recorded with a Canon EOS 600D digital single-lens reflex camera with a 55–250 mm lens (Canon Inc., Tokyo, Japan). The movements of the fish in each behavior test were tracked by using idTracker (version 2.1) and analyzed by using a computer with an Intel i7-5820K core at 3.3 GHz and with 64 GB RAM memory. The applied multiple behaviors test in adult zebrafish was performed according to a previous protocol [36,37]. Three replications were applied in these behavior tests.

### 2.7. Circadian Rhythm Locomotion Activity Test in Adult Zebrafish

After continuous tacrolimus exposure for 13 days, adult zebrafish in a group of four from each group were moved to the acrylic test tank (21 × 21 × 10 cm); this comprised ~2 L of water within an incubator to maintain the temperature at 26 ± 1 °C and a water pump to keep the water aerated. Lightboxes consisting of a light-emitting diode (LED) and an infrared light-emitting diode (IR-LED) were used as the light sources during the day and night cycles, respectively. The day cycle started at 09:00 and lasted until 22:59, while the night cycle started at 23:00 and lasted until 08:59. The fish behaviors were recorded every hour starting from 16:00 until 15:00 the following day. Each video was 1 min long and recorded by an infrared-sensitive charge-coupled device (detection window: 700–1000 nm) with a resolution of 1920 × 1080 pixels at 30 fps (3206_1080P module, Shenzhen, China). Later, the fish movements during the day cycle were tracked using idTracker, while UMAtracker (version 15) was used to track their movements in the night cycle [36,38]. The current test was based on a previous publication and performed in four replications [39].

### 2.8. Color Preference Assay in Adult Zebrafish

On the fifteenth day of exposure, the zebrafish in each group were put into an acrylic tank with dimensions of 21 × 21 × 10 cm and filled with ~1.5 L of water; the tank was divided into two sections to compare a set of two-color combinations from four colors (red, green, blue, and yellow) at once. Later, the fish movements were recorded for 30 min using a high-quality CCD camera (ONTOP, M2 module, Shenzhen, China) with a resolution of 3264 × 2448 pixels at a 30 fps frame rate. Finally, UMAtracker was used to analyze the position of the fish during the test to evaluate the fish color preferences between the two colors provided in each test [38]. The applied color preference assay was carried out in four replications and followed a method described previously [40].

### 2.9. Passive Avoidance Test (Short-Term Memory Test) in Adult Zebrafish

The exposed fish were subjected to the passive avoidance test on the final day of the exposure. This test was conducted in an acrylic tank with a size of 30 × 20 × 20 cm; this tank was equally divided into two compartments (black and white), and a movable separator was used to separate these compartments. Initially, the fish were individually placed in the tank without a separator for 5 min to recognize the novel environment. Later, during the training session, the fish were moved to the white compartment with the separator, not allowing them to swim to the black compartment for one minute. Then, the separator was taken out, allowing the fish to enter the dark compartment. However, when they entered the dark compartment, a mild electric shock (25 V, 1 mA) was administered. The shock was only given up to three times at 5 s intervals. If the fish did not move to the white compartment within 5 min of observation, they were put back in the white compartment. This session was repeated a maximum of three times, with one minute of acclimation in between. On the following day, the fish were placed again in the white compartment, and the time it took them to enter the dark compartment was counted as a short-term memory indicator. The test was conducted in four replications and the this test was performed according to a prior publication [41].

### 2.10. Statistical Analyses

All statistical analyses were conducted by using GraphPad Prism (GraphPad Software version 8 Inc., La Jolla, CA, USA), and the results from each test were evaluated with certain statistical tests, as described in each figure. All the graphs were also plotted by using GraphPad Prism and expressed either as mean or median. For the behavior test results, nonparametric tests were applied since the assumption of normal distribution was not required by these analyses [42].

### 2.11. Molecular Docking 1FKJ Human FKBP 12 in Complex with Tacrolimus

The atomic structure of fkbp12-fk506, an immunophilin immunosuppressant complex of *Homo sapiens* (Human) origin (PDB: 1FKJ), was extracted from the RCSB Protein Database with a resolution of 1.70 Å, which is within an acceptable range. The amino acids that bind the protein FKBP12 to the drug FK506, also known as tacrolimus, include TYR26, PHE36, ASP37, PHE46, GLU54, VAL55, ILE56, TRP59, ALA81, TYR82, HIS87, and ILE91 (Appendix A). The interactions involving ASP37, GLU54, ILE56, and TYR82 are of conventional hydrogen bonds, indicating a strong interaction within the extracted complex. The protein was prepared by removing excess molecules such as water molecules, chains, and ligands to prevent distortion. The ligand tacrolimus was removed from the complex and redocked to perform reverse molecular docking. The molecular docking results were docked using the PyRx software (https://pyrx.sourceforge.io/, accessed on 9 May 2023) using the Vina Wizard tool for docking and the OpenBabel tool for accessing the ligand and converting to a PDBQT format, making tacrolimus recognizable as a ligand. Reverse molecular docking was performed with the grid size described in Appendix A. The grid size was determined based on the known amino acid interactions of the extracted protein–ligand complex from the database.

## 3. Results

### 3.1. Acute Exposure to Tacrolimus Did Not Alter the Cardiac Rhythm and Physiology of Zebrafish Larvae

Here, the cardiotoxicity properties of tacrolimus after acute exposure were evaluated by assessing the cardiac physiology and heart rate of the fish through the ventral aorta posterior cardinal vein channel measurement of the larvae. From the results, no statistical differences were observed in all the cardiac physiology endpoints, including the stroke volume and cardiac output, which indicate that acute exposure to tacrolimus did not compromise the ability of the larval heart to pump blood (Figure 2A–D). Similarly, the same heart rate values between the control and the treated groups were also found in this assay (Figure 2G). Together with the normal levels of SD1 and SD2 in the ventricle chamber, these results signified the regularity of the atrium and chamber beats of the treated larvae (Figure 2E,F). To sum up, these findings highlight that an acute exposure to tacrolimus in the given concentrations did not produce cardiotoxicity in the zebrafish larvae.

### 3.2. Acute Exposure to Tacrolimus Did Not Affect the Blood Flow of Zebrafish Larvae

Since no alterations were observed in the cardiac rhythm and physiology of the larvae after being treated with tacrolimus, a blood flow test was also conducted to confirm whether tacrolimus caused any abnormalities in the larval cardiovascular performance. Here, the speed of the blood flow in the dorsal aorta of the larvae was evaluated. As expected, the treated groups in both concentrations exhibited a comparable blood flow to the control group. This phenomenon is indicated by the statistically similar level of average and the maximum blood flow velocity between the untreated and treated groups (Figure 3A–E). Taken together, these results suggest that acute exposure to tacrolimus in the given concentrations did not alter the cardiac rhythm and physiology of the zebrafish larvae, nor their blood flow.

### 3.3. Acute Exposure to Tacrolimus Did Not Alter Zebrafish Larval Respiratory Rate

Following the previous results regarding the cardiotoxicity of tacrolimus after acute exposure, it was intriguing to verify this result by measuring the oxygen consumed by the larvae post-tacrolimus exposure since there is a potential relationship between the breathing and heart rate in teleost fish [43]. After monitoring the oxygen level of each group for 75 min, it was found that the treated groups in both concentrations exhibited a statistically similar oxygen consumption to the untreated group, as shown in Figure 4A,B. Thus, these results confirmed that tacrolimus did not alter the larval respiratory rate, which is consistent with the results of the cardiotoxicity assessment.

### 3.4. Acute Exposure to Tacrolimus Increased the Zebrafish Larval Locomotion during the Light–Dark Locomotion Test

Research on fish has demonstrated a correlation between their swimming activity and heart rate and breathing rate, with these endpoints increasing during swimming and decreasing when swimming ceases [44]. Since no alterations were found in the heart and respiratory rates of the zebrafish larvae after being exposed to tacrolimus, a deeper study of their locomotion activity in the form of photomotor responses was essential in order to comprehensively evaluate the effect of tacrolimus on zebrafish larvae. Interestingly, after calculating their locomotion during the test, which included several light and dark transitions, hyperactivity was observed in the treated groups. During the light cycle, only the high-dose group displayed a statistically higher locomotion than the untreated group; meanwhile, during the dark cycle, hyperactivity was exhibited by both exposed groups (Figure 5A–C).

### 3.5. Acute Exposure to Tacrolimus Slightly Reduced the Habituation of Vibrational Startle Response (VSR) of Zebrafish Larvae

Since exposure to tacrolimus was shown to alter the locomotor activity of zebrafish larvae during the light–dark locomotion assay, it was possible that it might also impair their vibrational startle response. Therefore, a vibrational startle response assay (VSRA) was carried out to evaluate the response of the larvae to the vibrational stimuli generated by a tapping device. Interestingly, relatively normal habituation during the tapping stimulus was displayed by both treated groups, indicated by a similar total distance traveled during the occurrence of the tapping stimuli between the control and treated groups (Figure 6B). However, acute exposure to tacrolimus statistically increased the larval locomotion as a response to the first two given tapping stimuli (Figure 6A).

### 3.6. Chronic Exposure to Tacrolimus Reduced Locomotion and Altered Exploratory Behaviors of Zebrafish in the Novel Tank Test

Considering the minor acute effects of tacrolimus on the zebrafish larvae, the following studies were carried out to evaluate its toxicity toward zebrafish in an adult stage with the same concentrations but in a chronic exposure manner. After ~2 weeks of incubation with tacrolimus, the performance of the zebrafish in a new environment was evaluated by a novel tank test. By conducting this assay, several behavior endpoints were extracted, and each endpoint represents either the locomotor activity or exploratory behavior performances of the zebrafish during the test. From the results, low locomotion activity was observed in the fish treated with a high concentration of tacrolimus (100 ppb). This phenomenon was indicated by the lower average speed and rapid movement time of this group compared to the untreated group (Figure 7A,D). On the other hand, a normal locomotion level was displayed by the fish after being exposed to tacrolimus in a lower dose (1 ppb), shown by the similar levels of all locomotor activity endpoints between the tacrolimus-treated group and control group (Figure 7A–D). However, altered exploratory behaviors were detected in both treated groups, with significant behavioral changes observed in the high-concentration group. Although no statistical differences were found in some of the exploratory behavior endpoints, including the number of entries to the top and average distance to the center of the tank (Figure 7E,H), statistical deviations were calculated in other endpoints. Based on the differences found in the latency to enter the top, the total distance traveled at the top, and the time in top duration (Figure 7F,G,I), it was concluded that the tacrolimus-treated fish preferred to swim in the top portion of the tank compared to the untreated group. Taken together, while the high concentration of tacrolimus could reduce zebrafish locomotion, tacrolimus in both concentrations could alter the exploratory behaviors of zebrafish.

### 3.7. Chronic Exposure to Tacrolimus in a High Dose Resulted in Loosened Shoal Formation of Zebrafish

In addition to the novel tank test, various behavior tests were also conducted to comprehensively evaluate the behavior toxicity of tacrolimus in zebrafish. Interestingly, after being exposed to a high concentration of tacrolimus, the zebrafish formed a loosened shoal formation compared to the control group, indicated by the high average inter-fish distances, an average nearest neighbor distance, and an average farthest neighbor distance (Figure 8A,C,D). The effects of tacrolimus on the shoaling behaviors of zebrafish were similar to its effects on fish locomotion during the novel tank test, with only the high dose having prominent effects on behavior since no statistical differences in all shoaling behavior endpoints were found in the low-dose group (Figure 8A–D). However, tacrolimus did not affect the aggression, conspecific social interaction, and fear response behavior of zebrafish, since none of the behavior endpoints of both treated groups in those particular tests were statistically different from the untreated group (Appendix A).

### 3.8. Chronic Exposure to Tacrolimus Caused Some Changes in the Circadian Rhythm Locomotor Activity of Adult Zebrafish

Based on the novel tank test results, it was possible that the administration of tacrolimus could alter the locomotion of zebrafish in response to the novel environment; thus, it was intriguing to verify whether this alteration could affect their circadian rhythm locomotor activity. Therefore, the circadian rhythm locomotor activity was assessed by evaluating the fish locomotion within a one-minute observation time with one-hour time intervals for 24 h. Based on the results, although no statistical changes were displayed in terms of locomotion, changes in their movement orientation were displayed by the treated groups during the day cycle (Figure 9C,E–G). This phenomenon was indicated by the high level of meandering, which may indicate a zigzag-like movement possessed by the exposed groups (Figure 9D). Meanwhile, during the night cycle, alterations in locomotion were only observed in the low-concentration groups, as shown in the average speed, freezing, swimming, and rapid movement ratios. This hyperactivity might also elucidate the statistically low level of meandering, although no changes were shown in the average angular velocity (Figure 9I,J).

### 3.9. Chronic Exposure to Tacrolimus Induced Short-Term Memory Loss in Adult Zebrafish

A previous study in a murine model found that tacrolimus-treated mice showed significantly decreased hippocampal-dependent spatial learning and memory function [45]. Therefore, a short-term memory test was conducted to evaluate whether chronic exposure to tacrolimus could also impair the memory function of the zebrafish. From the results, the untreated group showed a higher latency to enter the dark chamber at 24 h after training. However, the treated groups displayed different results, with a similar latency between before and after training, indicating that the treated fish exhibited significantly lower memory retention (Figure 10). Taken together, these results demonstrate that chronic exposure to tacrolimus in relatively low doses could induce short-term memory loss in zebrafish.

### 3.10. Chronic Exposure to Tacrolimus Altered the Color Preferences of Zebrafish

Considering the abundance of zebrafish-specific photoreceptors in their cone cells, which are used to distinguish color, it is intriguing to also evaluate whether chronic exposure to tacrolimus in the given concentrations could cause changes in their color preferences. Generally, adult zebrafish have a clear color preference, ranked as follows: red > blue > green > yellow. Although no statistical differences were found in the majority of the color choice index, some differences were still observed in some color combinations (Figure 11A,C–E). First, in the green–yellow color combination, the low-concentration group showed a statistically lower level of green color choice and a statistically higher level of yellow color choice than the untreated group (Figure 11B). Furthermore, tacrolimus exposure in low concentrations also resulted in color choice differences during the test when using a red–yellow color combination, as this treated group displayed a statistically higher yellow color choice index than the control group. In addition, a similar color preference alteration in the red–yellow color combination was also observed in the high-concentration group, although it was not as severe as in the low-dose group (Figure 11F). To sum up, acute exposure to tacrolimus, especially in a relatively low dose, could impair the color preferences of zebrafish in some color combinations.

### 3.11. FKJ Human FKBP12 in Complex with Tacrolimus

The preliminary results generated a pose with binding energies ranging from −5.8 kcal/mol to −4.0 kcal/mol (Appendix A); while the resulting amino acid interactions of every binding conformation are limited, it is noteworthy that none of the binding poses resulted in unfavorable interactions. However, the RMSD of the redocked protein–ligand complex was not determined due to the unlikely superimposition pose, which is more than 2 Å. The superimposition suggests that redocking may have manipulated the positioning of the ligand to the protein. While the docking site or area is the same, the head or tail of the ligand/compound might have inversely positioned itself in contrast to the initial extracted complex conformation.

## 4. Discussion

This is the first study to evaluate the toxicities of tacrolimus in relatively low doses to zebrafish in both larval and adult stages. Based on the results for the larvae, acute exposure did not cause any significant alteration in their cardiac rhythm and physiology, blood flow, and respiratory rate; however, high locomotion was demonstrated by the treated fish larvae in response to light changes and vibration stimuli. On the other hand, chronic exposure to tacrolimus resulted in lower locomotor activities and abnormal exploratory behaviors in adult zebrafish during the novel tank test. Furthermore, although no significant behavior alterations were found in terms of the aggressiveness, fear responses, and conspecific social interaction of the fish after tacrolimus treatment, loosened shoal formation was observed in the treated group, together with a slightly disrupted circadian rhythm locomotion, especially in the low-dose group. In addition, it was also found that tacrolimus exposure in the given concentrations could mildly change the color preference of the zebrafish. Finally, as the most important finding in this study, chronic exposure to this compound resulted in short-term memory loss in zebrafish. To sum up, although different doses of tacrolimus were not found to have toxic effects on every aspect of the zebrafish tested in the present study, this study highlighted the necessity of taking precaution prior to the usage of tacrolimus, especially in aquatic animals, even at relatively low concentrations since it might still exert some adverse effects on animals.

### 4.1. Tacrolimus in Affecting Locomotion in Zebrafish Larvae

In zebrafish, locomotion is a complex behavior produced by the activity of various neurons in the neurotransmitter systems that are located in brain areas and/or specific circuits and are also common to other vertebrates [46,47]. As an act crucial for the survival of animals, the locomotion analysis in zebrafish larvae could provide accurate information regarding the potency of the neuroactive properties of compounds [48]. The light–dark locomotion test is one of the locomotion analyses that analyzes the zebrafish larval photomotor response by evaluating their locomotion activity and movement pattern during alternating light and dark conditions following a period of acclimatization. Generally, zebrafish larvae show increased locomotor activity in the light–dark transition, with decreased locomotor activity in the dark–light transition [49]. Here, while the treated groups displayed a relatively normal pattern of photomotor response, with higher locomotor activity in the dark cycle than light cycle, they exhibited higher locomotion in both cycles, especially the high-dose group, signifying hyperactivity-like behaviors. This behavior alteration might indicate the neuro-stimulant properties of tacrolimus, which are similar to the effect of adrenaline presented in a prior study. In addition, they also found an aggravated startle response of zebrafish larvae after adrenaline treatment; this was also observed in the tacrolimus-treated larvae during the VSRA in the present study, although the increased activity was only observed during the initial period when vibration stimuli were given [49]. Furthermore, the current results are somewhat in line with the results of previous studies finding that FK506 treatments at 1 µM increased the activity of larval zebrafish and decreased their optomotor responses. However, they demonstrated that treatment with the DYRK inhibitor proINDY induced opposite effects, which is consistent with models of calcineurin-NFAT signaling. Meanwhile, they also demonstrated that FK506 treatment led to an increase in excitability compared to the controls during the acoustic startle responses test [8]. Interestingly, in rats, while tacrolimus was demonstrated to have significant beneficial effects on rats with various treatments and injuries, such as arthritis and spinal cord injury [50,51], especially in the process of recovering their locomotion, tacrolimus alone had no significant effect on the basal locomotor activity of the normal rats, at least in the given concentrations tested in the study [52,53]; this indicates the differences in the effects that tacrolimus has on animal locomotion between rats and fish, especially zebrafish. Taken together, tacrolimus in the given concentrations has harmful effects on the activity and visually guided behaviors of zebrafish larvae, which may be related to the calcineurin signaling blockage caused by this compound.

### 4.2. High Dose of Tacrolimus in Affecting the Behavior Performance in a Novel Environment and Shoaling Formation of Adult Zebrafish

On the other hand, varied behavior results were found in the adult zebrafish. Interestingly, the low dose of tacrolimus resulted in different behaviors compared to the fish that were exposed to 100 ppb of tacrolimus. Here, the fish treated with a higher dose of tacrolimus exhibited lower locomotion compared to the control group, while reduced exploration behaviors were observed in both groups treated during the novel tank test. The novel tank test is one of the fish behavior tests commonly utilized to measure the anxiety level of fish by assessing the fish’s response to unfamiliar situations or stimuli. Generally, when placed in a novel tank, zebrafish initially exhibit pronounced behavioral inhibition, with gradual habituation after prolonged exposure [54,55]. Abnormalities in the behavior endpoints can be indicators of anxiety-like behaviors in zebrafish in a way comparable to the rodent open-field test. Based on a previous study, spending more time in the top portion of the tank, as observed in the current study, is usually related to a low anxiety level [56]. Although the treated fish also exhibited a slight decrement in their average speed and rapid movement time, which usually indicates a higher anxiety level, this does not seem to apply in the current results since a prior study demonstrated that exposure to fluoxetine, an antidepressant with anxiolytic effect, and lysergic acid diethylamide (LSD), a hallucinogenic drug, decreased the locomotor activity of mosquitofish (*Gambusia holbrooki*) and zebrafish, respectively [57,58]. In addition, the potential anxiolytic effects of a high dose of tacrolimus were also exhibited in the shoaling test results, which showed that loosened shoals were formed by the treated fish. These results are similar to previous findings that demonstrated the anxiolytic effect of PAH benzo[a]pyrene, a polycyclic aromatic hydrocarbon, in altering the shoal dynamics and decreasing the locomotion of zebrafish [59]. Alterations in shoaling behavior and a decrement in shoal cohesion were also observed after the administration of chlordiazepoxide, an anxiolytic drug, to zebrafish. In addition, after investigating the sensitivity of several behavioral paradigms measuring zebrafish anxiety-like behavior, their study found that the novel tank and shoaling tests were the most sensitive to anxiolytic effects, as supported by another study, suggesting the reasons for the absence of significant differences in the other behavior tests conducted in the present study; these were the aggressiveness, fear response, and conspecific social interaction test [60,61]. Nevertheless, these results suggest that chronic exposure to a high dose of tacrolimus in adult zebrafish results in a slight abnormality in the habituation response of zebrafish to a novel environment and shoal size, which indicates the potential anxiolytic effect of tacrolimus.

### 4.3. Low Dose of Tacrolimus in Affecting the Behavior Performance in a Novel Environment and Circadian Rhythm Locomotor Activity of Adult Zebrafish

On the contrary, some behavior outcomes of the tacrolimus-treated group at a lower dose indicate that it has potential anxiogenic effects on adult zebrafish. In the novel tank test, this treated group displayed exploratory behaviors that were similar to the high-dose group, especially in terms of a preference for the upper portion of the tank, which might be due to the slightly high value of the number of entries to the top portion of the tank and locomotion; this is indicated by the calculated locomotor activity endpoints, although they did not reach a statistical difference compared to the control group. Therefore, we hypothesized that the slightly abnormal exploratory behaviors were related to the relatively high locomotion caused by the low-dose tacrolimus, causing the fish to transition to the top and bottom half of the tank more frequently, indicating the potential anxiogenic effects of tacrolimus, since increased locomotion in fish might also act as a signal of increased anxiety-like behavior; this is typically exhibited as avoidance or escape behavior and an increase or surge in autonomic activity [60,62]. Alterations in exploratory behaviors might affect the survival or fitness of fish, since well-developed exploratory behavior may provide benefits during a life stage in which animals shift into a new habitat or disperse over a larger area if the mortality risk in the present habitat is increasing and/or the prey abundance decreases [63,64]. Furthermore, this hypothesis was formulated by also considering the disruptions in the circadian rhythm locomotor activity of the fish after exposure to tacrolimus at a low concentration. In humans and animals, it is already well known that the disturbance of circadian clocks is particularly associated with the development of mood and anxiety disorders [65,66,67,68]. In addition to human data, many studies have demonstrated the link between circadian rhythms and psychiatric endophenotypes. As an example, *cryptochrome*-deficient mice that were unable to express endogenous circadian rhythms were demonstrated to have a pronounced anxiety-like phenotype, which was manifested in abnormalities at the cognitive and social behavioral levels [69]. Similarly, in aquatic organisms, including fish, their vulnerability to the effects of pharmaceutical pollution may also cause circadian inactivity rhythms, which could impair their cognitive performance and swimming behavior [70,71,72]. For instance, a prior study in adult zebrafish demonstrated that chronic exposure to butyl-paraben (BuP), one of the parabens already shown to induce anxiety-like behavior in zebrafish larvae, results in circadian rhythm modulation and abnormal neurobehavioral responses to light stimulation; they later found that the explainable cause of this phenomenon is the ability of BuP to cross the blood–brain barrier and modulate the level of transcripts that are associated with phototransduction and the circadian rhythm [73,74]. Taken together, chronic exposure to tacrolimus might cause relatively minor behavior alterations in adult zebrafish in a dose-dependent manner; this requires further investigation in future studies.

### 4.4. Tacrolimus in Altering the Color Performance of Adult Zebrafish

Next, despite the slight differences observed in the behavior of the groups exposed to low and high concentrations of tacrolimus mentioned above, both treated groups experienced minor dysregulations in their color preference. Environmental colors are one of the important aspects of rearing animals, especially fish, since they may act as a stress buffer or even a source of stress [75,76]. Generally, species may possess their own color preferences; for example, zebrafish, which are known to have a highly developed visual system that is comparable to that of humans, prefer blue-colored zones over yellow-colored zones [77,78]. However, stressful events may alter their perceptions and reactions to the physical environment, including their spontaneous color preferences, as shown in a prior study on rainbow trout (*Oncorhynchus mykiss*) [79]. In their study, air exposure, which was used as a stressor, affected the color preference of rainbow trout, suggesting that abiotic stressors may also affect the color perception or behavior plasticity of fish [80]. Meanwhile, other studies in adult zebrafish have also observed the significant absence of color discrimination and color preference after the administration of ethanol and acrylamide, a substance with known neurotoxic effects, respectively [77,81]. In addition to the consequences of structural changes in the retina of zebrafish, the later study also suggested that the changes might be due to direct damage to the motor circuitry of zebrafish. Thus, further work is required to investigate any sensory deficits in the fish after being treated with tacrolimus. Nevertheless, chronic exposure to tacrolimus in the given concentrations resulted in minor changes in the color preference of the zebrafish.

### 4.5. Tacrolimus in Compromising the Cognitive Performance of Adult Zebrafish

Interestingly, all the treated groups also displayed short-term memory loss during the passive avoidance tests. Normally, zebrafish display robust memory and learning abilities, exhibiting strong habituation responses and good learning in tasks that use various stimuli, including food, making them a suitable animal model to use when studying neurodegenerative diseases [82,83]. However, their memory and learning ability could be strongly affected by stress, which might come in many ways, including drug exposure. Unlike in zebrafish, many studies have evaluated the effect of tacrolimus on the memory and cognitive performance of rats. Interestingly, despite the recognition of its neurologic toxicity, including its impact on speech disorders, some studies have indicated the beneficial effects of tacrolimus [84]. For example, tacrolimus was demonstrated to reduce cognitive impairment and exhibit a memory-enhancing action in cerebral ischemia rats and alcohol-treated rats, respectively [85,86]. However, other studies have reported diverse central nervous system side effects, including memory impairment, in more than 10% of animals clinically prescribed tacrolimus [87,88]. Furthermore, another report in mice showed that tacrolimus caused cognitive function impairments via synaptic imbalance and the inducement of oxidative stress in the hippocampus, similar to the current results [45]. Moreover, Hadjiasgary et al. also demonstrated that injections of tacrolimus impaired the acquisition of inhibitory avoidance learning and later suggested that tacrolimus selectively interferes with the acquisition, retention, and retrieval of information processing in CA1, one of the proteins with the highest density in the hippocampus [89]. Taken together, the observed neurotoxic effects of tacrolimus in the present study presumably resulted from its ability to cross the blood–brain barrier and affect cerebral function, as well as affect the neurochemical system, potentially resulting in stress; this could eventually impair fish memory, as much experimental evidence has indicated the link between stress and memory [90].

### 4.6. The Binding Energies of Tacrolimus against Human FKB12

Finally, due to the well-known ability of tacrolimus to bind to immunophilins that exhibit peptidylprolyl cis/trans isomerase (PPIase) activity, molecular docking was conducted to calculate the binding energies of tacrolimus (FK506) against FKBP12, one of the canonical FK506-binding protein (FKBP) members, in order to elucidate its toxicity mechanism toward zebrafish [91,92]. Unfortunately, since the homolog of this protein was not available in the zebrafish database, the current study used the atomic structure of this protein from humans. Generally, FKBPs with various molecular weights are the principal intracellular targets for tacrolimus [93]. After being formed with FKBPs, the FKBP/FK506 complexes inhibit not only PPIase activity, but also the phosphatase activity of the secondary target calcineurin (CaN); this prevents the dephosphorylation of NF-AT that is required for IL-2 gene expression and T-cell activation [94,95]. Here, we found that the binding affinity ranged from −5.8 kcal/mol to −4.0 kcal/mol, which could be considered as not particularly high according to the standard of drugs (−6.00 kcal/mol) [96]. However, despite the values, these results still indicate the potential activity of tacrolimus against human FKBP12 and might help in elucidating its mechanism in altering the fish behaviors observed in the present study, since a prior study found that immunophilins are more abundant in nervous tissues than in immune tissues, indicating their significance in neurons [97,98]. Nevertheless, while the binding properties of tacrolimus to FKB12 might partly play a role in the observed behavioral alterations, future studies still need to be conducted to fully comprehend the toxicity mechanism of tacrolimus and verify the current results by using data from zebrafish.

## 5. Conclusions

In conclusion, the current study demonstrated the toxicity of tacrolimus in relatively low concentrations to zebrafish, especially in their behaviors, social cohesion, and cognitive performances; this was observed in both the larval and adult stages after acute and chronic exposures, respectively, with some alterations observed in a dose-dependent manner (Figure 12). We hypothesized that tacrolimus may possess anxiotropic properties, which make it capable of modifying anxiety in fish, causing minor alterations in some of their behaviors. These properties might have resulted from the ability of tacrolimus to cross the blood–brain barrier and affect the nervous system of fish, and also alter the short-term memory of fish. Afterward, based on the molecular docking results obtained for tacrolimus and human FKBP12, it is suggested that this interaction might partly contribute to the observed alterations. Nevertheless, a further analysis of the mechanism implicated in the amelioration of the pharmacologic activity of tacrolimus is still required in the future, and considering the potential toxicity of tacrolimus, precautions should be taken and its adverse effects closely monitored before it is used and clinically applied. Finally, the present work would greatly benefit clinicians and researchers and allow them to further understand the toxicity of tacrolimus to zebrafish.

## Figures and Tables

**Figure 1 biology-13-00112-f001:**
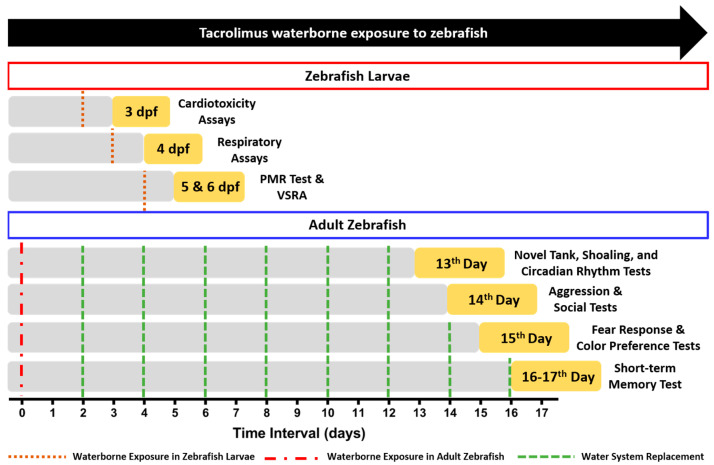
The detailed experiment protocols regarding the tacrolimus exposure and toxicity tests’ schedule in the present study. For the waterborne exposure of tacrolimus in the zebrafish larvae, the brown line indicates the start of the exposure (day 0). For the waterborne exposure of tacrolimus in the adult zebrafish, the start of the exposure (day 0) is indicated with the red line while the green lines show the times that the water replacement of the exposure solution was conducted every two days.

**Figure 2 biology-13-00112-f002:**
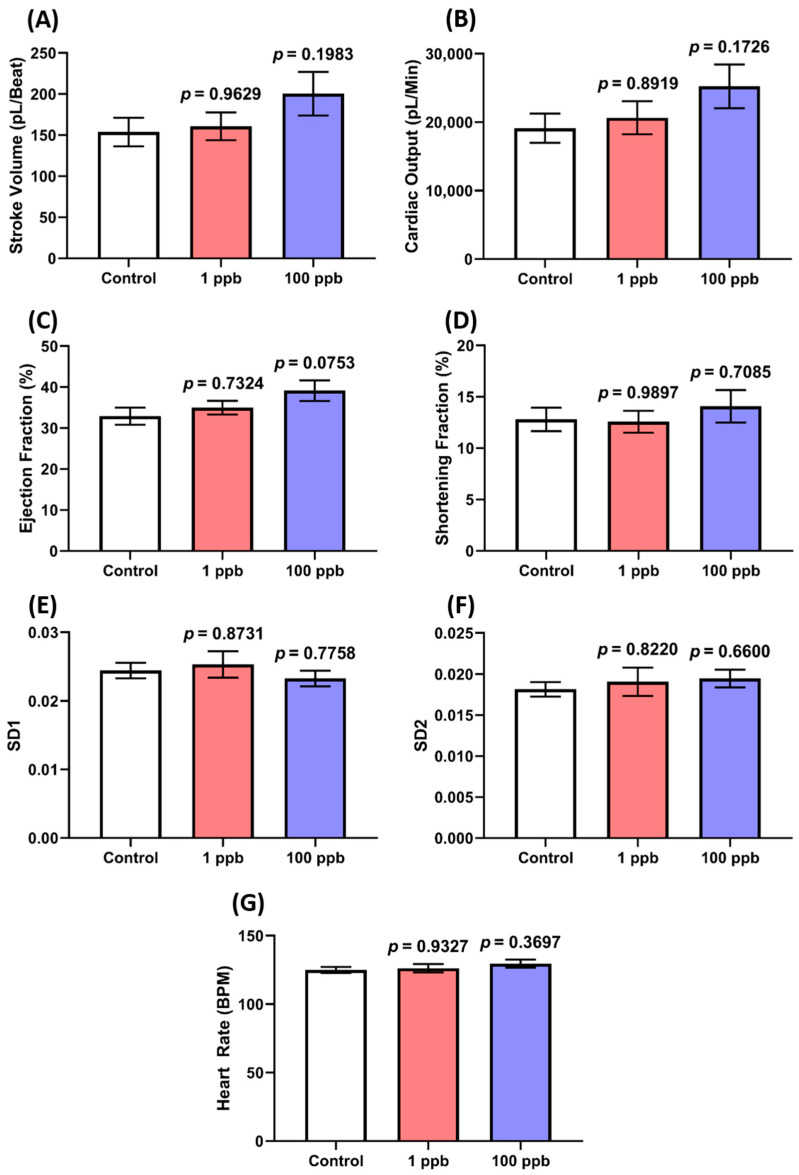
Comparison of several cardiac physiology endpoints: (**A**) stroke volume, (**B**) cardiac output, (**C**) ejection fraction, (**D**) shortening fraction, heart rate variability in (**E**) SD1 and (**F**) SD2 of ventricle chamber, and (**G**) heart rate of ventricle chamber in three dpf zebrafish larvae after exposure to 0 (control), 1 ppb, and 100 ppb of tacrolimus for 1 day. The data were analyzed by using a one-way ANOVA test, followed by Dunnett’s multiple comparisons test, and are presented as mean with SEM. The statistical comparisons were performed between the control and each treated group (*n* control and 100 ppb = 28, *n* 1 ppb = 24).

**Figure 3 biology-13-00112-f003:**
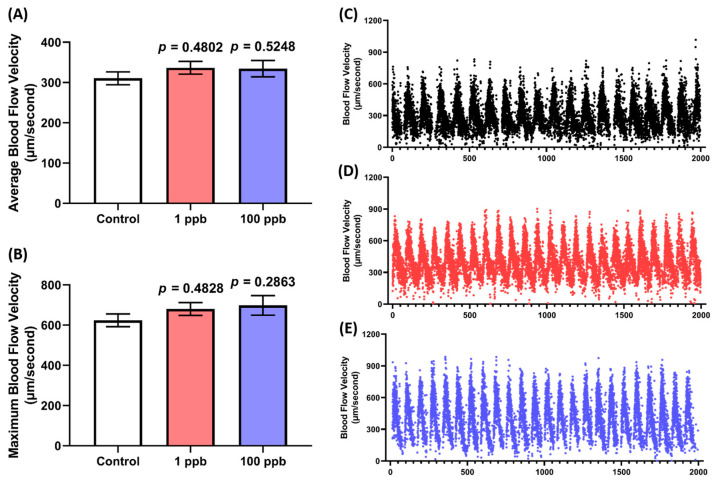
Comparison of (**A**) average and (**B**) maximum blood flow speed in zebrafish larvae after exposure to 0 (control), 1, and 100 ppb of tacrolimus. The data were analyzed by using a one-way ANOVA test, followed by Dunnett’s multiple comparisons test, and are presented as box and whiskers (min to max). The statistical comparisons were performed between the control and each treated group (*n* control = 30, *n* 1 ppb = 27, *n* 100 ppb = 29). The time chronology of the monitoring of the blood flow velocity in the dorsal aorta of a single representative zebrafish larva from (**C**) 0 (control), (**D**) 1, and (**E**) 100 ppb of the tacrolimus-treated groups.

**Figure 4 biology-13-00112-f004:**
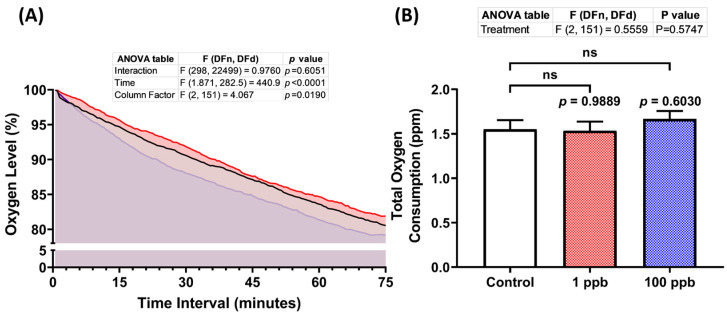
(**A**) Oxygen consumption level per minute in four dpf zebrafish larvae after 1 day of exposure to 0 (control), 1 ppb, and 100 ppb of tacrolimus during the respiratory rate assay. The data were analyzed using a two-way ANOVA test with Geisser–Greenhouse correction and expressed as mean. (**B**) Comparison of total oxygen consumption of the tested zebrafish larvae. The data were analyzed by using one-way ANOVA, followed with Dunnett’s multiple comparisons test, and are presented as mean with SEM. The statistical comparisons were performed between the control and each treated group (*n* control and 100 ppb = 53, *n* 1 ppb = 48; ns = not significant).

**Figure 5 biology-13-00112-f005:**
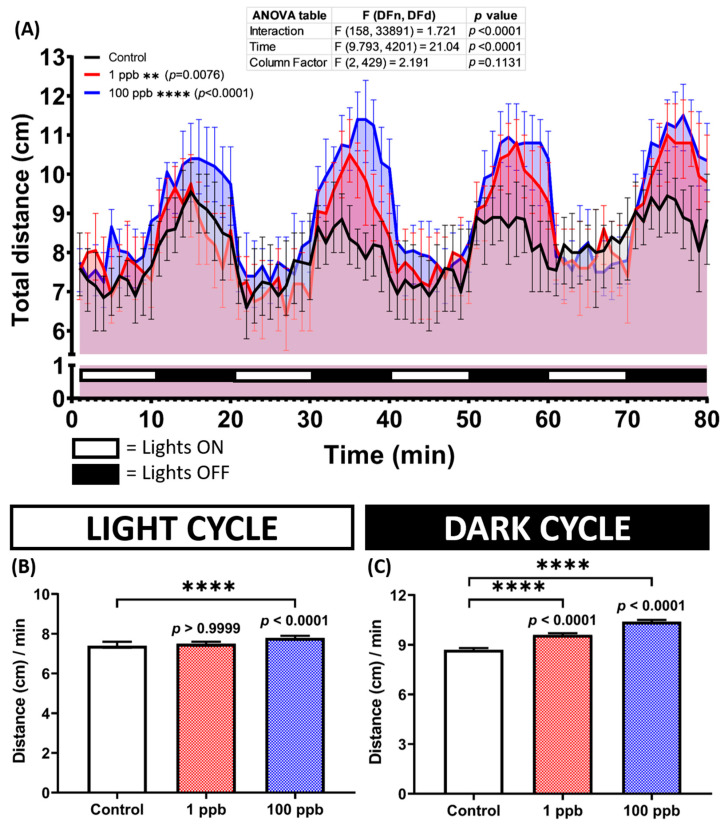
(**A**) Total distance traveled per minute by five dpf zebrafish larvae after 1 day of exposure to 0 (control), 1 ppb, and 100 ppb of tacrolimus during both light and dark cycles. The data were analyzed by using a two-way ANOVA test with Geisser–Greenhouse correction, followed by Dunnett’s multiple comparisons test. (**B**,**C**) Comparison of the total distance traveled by the tested zebrafish larvae in light and dark cycles, respectively. The data were analyzed using the Kruskal–Wallis test, followed by Dunn’s multiple comparisons test. The statistical comparisons were performed between the control and each treated group. All data are expressed in the median with 95% CI (*n* = 96; ** *p* < 0.01; **** *p* < 0.0001).

**Figure 6 biology-13-00112-f006:**
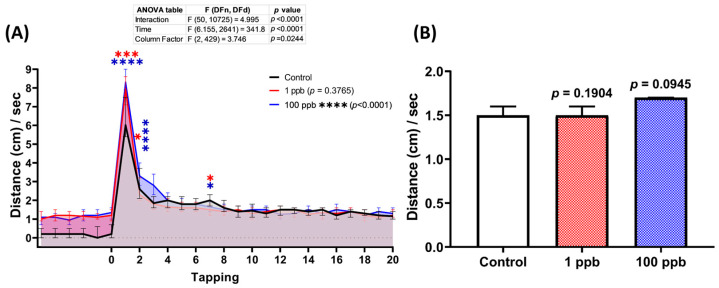
(**A**) Total distance traveled per second by six dpf zebrafish larvae after 2 days of exposure to 0 (control), 1 ppb, and 100 ppb of tacrolimus during the vibrational startle response assay. The data were analyzed using a two-way ANOVA test with Geisser–Greenhouse correction, followed by Dunnett’s multiple comparisons test. (**B**) Comparison of the average total distance traveled by the tested zebrafish larvae during the tapping stimuli. The data were analyzed using the Kruskal–Wallis test, followed by Dunn’s multiple comparisons test. The statistical comparisons were performed between the control and each treated group. All data are expressed as the median with 95% CI (*n* = 96; * *p* < 0.05, *** *p* < 0.001, **** *p* < 0.0001).

**Figure 7 biology-13-00112-f007:**
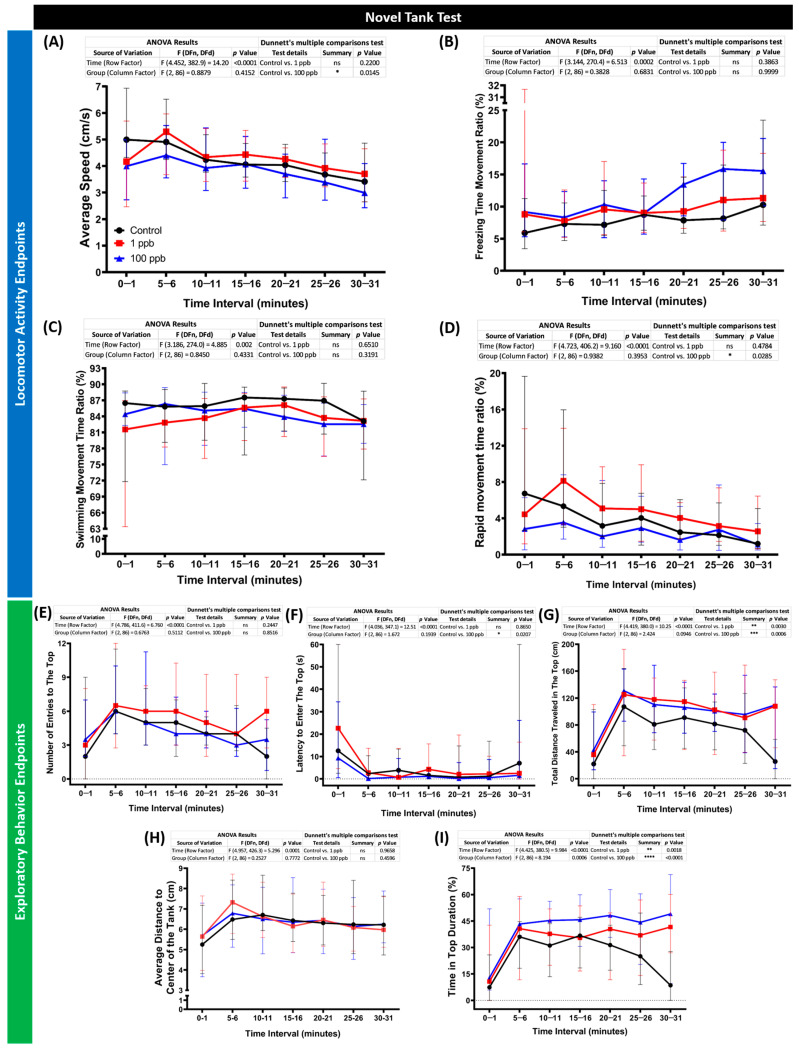
(**A**) Average speed, (**B**) freezing time movement ratio, (**C**) swimming movement time ratio, (**D**) rapid movement time ratio, (**E**) number of entries to the top, (**F**) latency to enter the top, (**G**) total distance traveled in the top, (**H**) average distance to center of the tank, and (**I**) time in top duration behavior endpoints of zebrafish after being exposed to tacrolimus at two different concentrations; 1 ppb (red) and 100 ppb (blue) compared to control (black). The data are expressed as the median with an interquartile range. The statistical analyses were conducted by two-way ANOVA with Geisser–Greenhouse correction. To observe the main column (tacrolimus) effect, Dunnett’s multiple comparison test was carried out. The statistical comparisons were performed between the control and each treated group (*n* control = 29, *n* tacrolimus = 30; * *p* < 0.05, ** *p* < 0.01, *** *p* < 0.001, **** *p* < 0.0001).

**Figure 8 biology-13-00112-f008:**
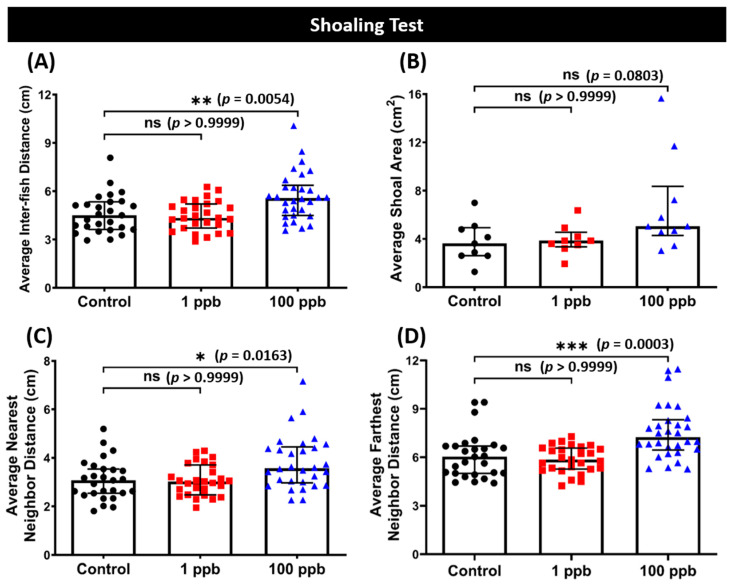
(**A**) Average inter-fish distance, (**B**) average shoal area, (**C**) average nearest neighbor distance, and (**D**) average farthest neighbor distance of zebrafish after being exposed to tacrolimus at two different concentrations: 1 ppb (red) and 100 ppb (blue) compared to control (black). The data are expressed as the median with an interquartile range. The statistical analyses were conducted using the Kruskal–Wallis test, followed by Dunn’s multiple comparisons test. The statistical comparisons were performed between the control and each treated group (shoal size = 3 fishes; *n* control & 1 ppb = 27, *n* 100 ppb = 30; ns = not significant, * *p* < 0.05, ** *p* < 0.01, *** *p* < 0.001).

**Figure 9 biology-13-00112-f009:**
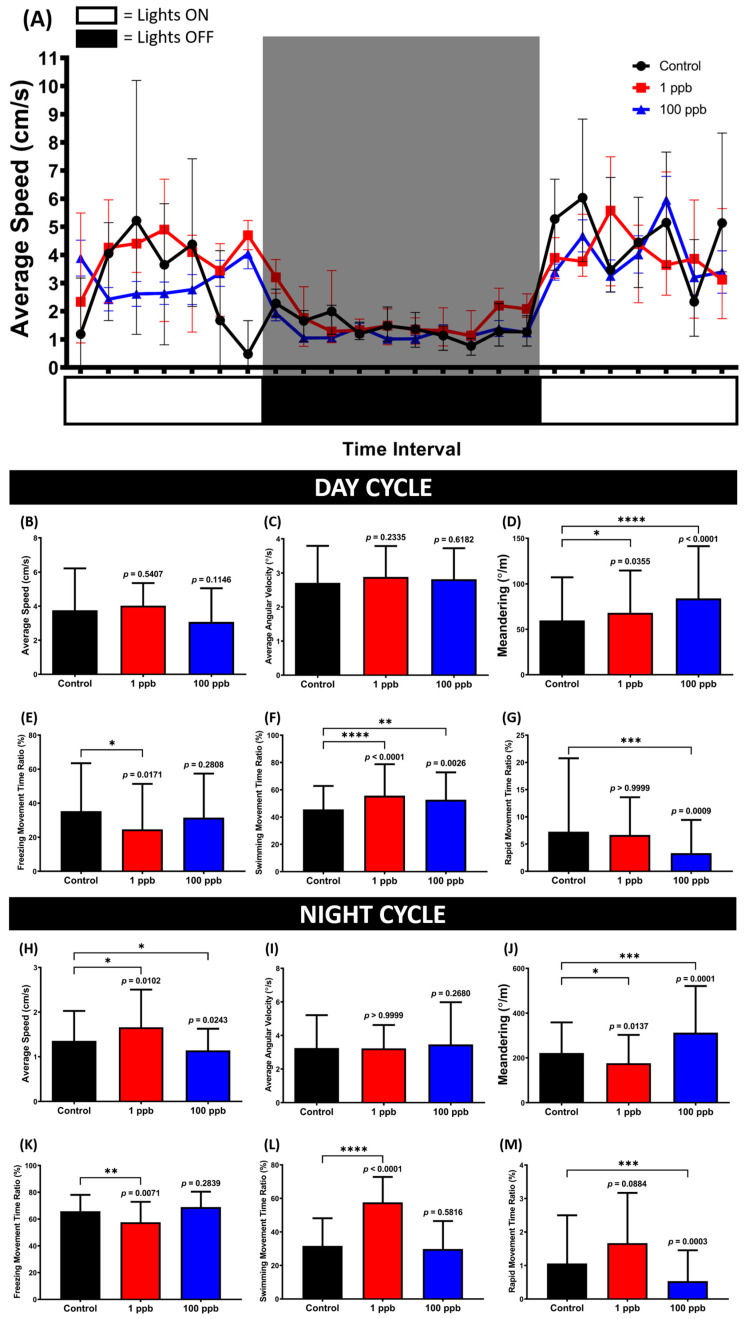
(**A**) Locomotor activity patterns of zebrafish that were chronically exposed to 0 (control), 1 ppb, and 100 ppb of tacrolimus during 24 h of observation. The data are expressed as mean with SEM. (**B**,**H**) Average speed, (**C**,**I**) average angular velocity, (**D**,**J**) meandering, (**E**,**K**) freezing movement time ratio, (**F**,**L**) swimming movement time ratio, and (**G**,**M**) rapid movement ratio of the tested fish during the day and night cycles, respectively. The data are expressed as the median with interquartile range and were analyzed using the Kruskal–Wallis test, followed by Dunn’s multiple comparisons test. The statistical comparisons were performed between the control and each treated group (*n* control and 1 ppb = 12, *n* 100 ppb = 8; * *p* < 0.05, ** *p* < 0.01, *** *p* < 0.001, **** *p* < 0.0001).

**Figure 10 biology-13-00112-f010:**
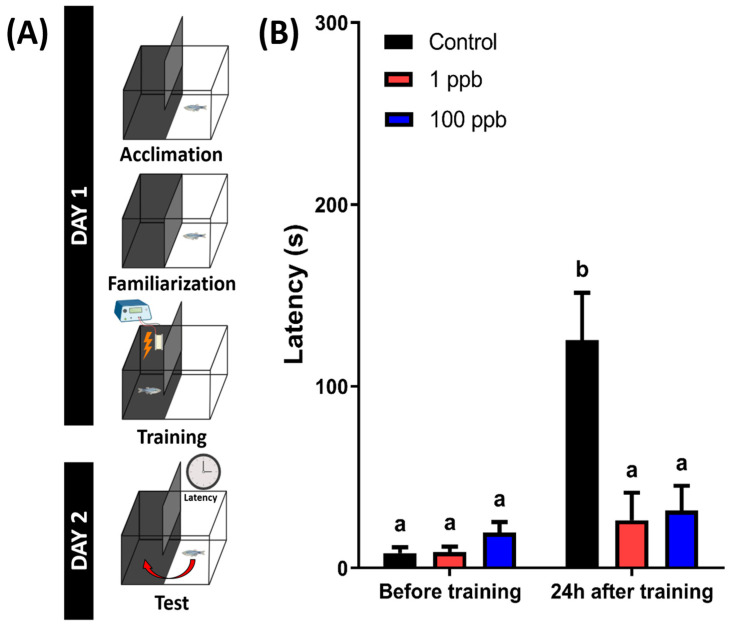
(**A**) The schematic of the passive avoidance experimental protocol (created with BioRender.com) and (**B**) the latency of fish chronically exposed to 0 (control), 1 ppb, and 100 ppb of tacrolimus to swim into the dark chamber 24 h after training sessions (right). The data are presented as mean with SEM and were analyzed with two-way ANOVA followed by Tukey’s multiple comparison test. Different letters signify statistical differences (*p* < 0.05) (*n* control = 18, *n* 1 ppb = 14, and 100 ppb = 16).

**Figure 11 biology-13-00112-f011:**
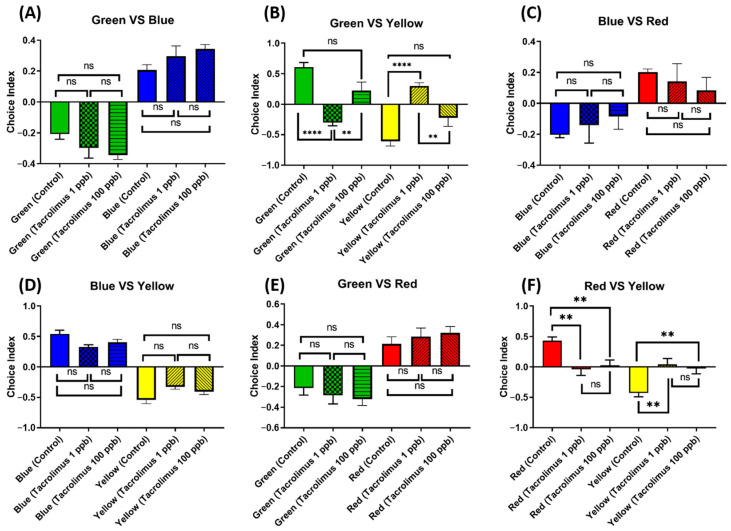
Comparison of the color preference behavior of zebrafish after being chronically treated with 0 (control), 1, and 100 ppb of tacrolimus. The four color combinations were as follows: (**A**) green–blue, (**B**) green–yellow, (**C**) blue–red, (**D**) blue–yellow, (**E**) green–red, and (**F**) red–yellow. The data are expressed as the mean ± SEM values and were analyzed by one-way ANOVA (*n* control = 9, *n* 1 and 100 ppb = 11; ** *p* < 0.01, **** *p* < 0.0001).

**Figure 12 biology-13-00112-f012:**
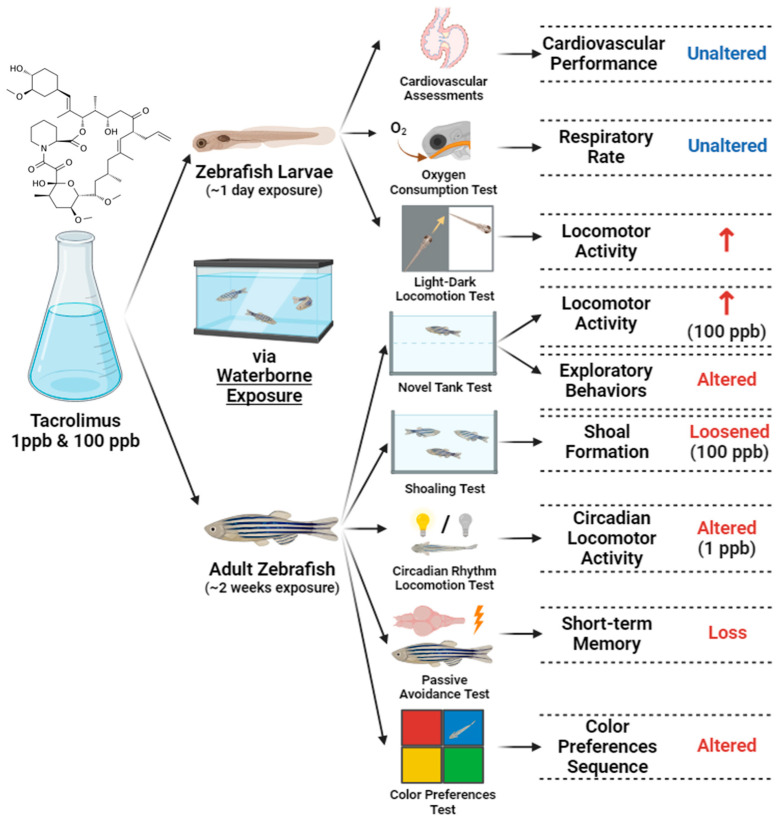
Summary of the toxicities of tacrolimus to larval and adult zebrafish observed in the current study after acute and chronic administration via waterborne exposure (↑: increased; created with BioRender.com).

## Data Availability

The dataset is available upon request from the authors.

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
