# Peer review of "Evaluation of Tacrolimus’ Adverse Effects on Zebrafish in Larval and Adult Stages by Using Multiple Physiological and Behavioral Endpoints"

_biology, 2024, doi:10.3390/biology13020112_

Round 1

Reviewer 1 Report

Comments and Suggestions for Authors

1)        Introduction should be shortened,  or  detailed analysis  moved  to discussion

2)        Behavioral assays are lacking  in details, especially shoaling and  should be separated

3)        Adult aquarium vendor  needs  to be identified, and  if  possible  identify  fish  source

4)        Stats  in figures  need  to be  clarified;  Are  the  comparison against control?  If  no dose- difference is  observed to be significant, specify so in legend.

5)        The  light-dark  a test of  locomotion ?  Please  discuss  this  further

6)        The authors  may want to subdivide the discussion as  to facilitate readership

Author Response

Comments and Suggestions for Authors

  • Introduction should be shortened, or detailed analysis moved to discussion

Thank you for the suggestion. The authors tried their best to shorten the introduction section by removing unessential parts, including the application performance of tacrolimus compared to other chemicals and some of the detailed analysis results from previous studies.

  • Behavioral assays are lacking in details, especially shoaling and should be separated

The authors appreciated the comment. In the updated version of the manuscript, more detailed information regarding the multiple behaviors assay, especially the shoaling test, was added to the Materials and Methods section as the reviewer suggested. In addition, actually, these behavior tests were based on our previous publication which provides more information regarding the tests if the readers need more detail regarding the behavior tests, however, the authors admitted that this matter was not clearly highlighted in the text, therefore, changes were made to highlight this issue.

  • Adult aquarium vendor needs to be identified, and if possible identify fish source

Thank you for the reminder. Additional information regarding the adult aquarium vendor, which was Zgenebio Inc. in Taipei, Taiwan (https://www.zgenebio.com.tw/english.html), was added to the manuscript. However, the fish source information could not be included in the text since it was not provided by the vendor.

  • Stats  in figures  need  to be  clarified;  Are  the  comparison against control?  If  no dose- difference is  observed to be significant, specify so in legend.

The authors thanked the reviewer for the questions. It is true that some statistical test information in some figures was unclear, thus, these issues needed to be clarified. All of the statistical tests done in Figure 2 to Figure 9 compare each treated group with the control group. Therefore, information regarding this matter was added to their respective figure legend.

  • The  light-dark  a test of  locomotion ?  Please discuss this further

Thank you for the question. Generally, a light-dark locomotion test in zebrafish larvae is utilized to evaluate the photomotor responses of larvae by measuring their locomotor activity and movement pattern with the changes in light intensity. However, since the treated groups exhibited a similar pattern of locomotion to the control group during the test, which indicated relatively normal photomotor responses, the high level of average speed shown by the treated larvae might be due to the high locomotion possessed by the treated larvae itself as some studies mentioned below also demonstrated the functionality of this test in evaluating larval locomotion.

Basnet, R. M., Zizioli, D., Taweedet, S., Finazzi, D., & Memo, M. (2019). Zebrafish larvae as a behavioral model in neuropharmacology. Biomedicines, 7(1), 23.

Lindemann, N., Kalix, L., Possiel, J., Stasch, R., Kusian, T., Köster, R. W., & von Trotha, J. W. (2022). A comparative analysis of Danionella cerebrum and zebrafish (Danio rerio) larval locomotor activity in a light-dark test. Frontiers in Behavioral Neuroscience, 16, 885775.

Peng, X., Lin, J., Zhu, Y., Liu, X., Zhang, Y., Ji, Y., ... & Li, Q. (2016). Anxiety-related behavioral responses of pentylenetetrazole-treated zebrafish larvae to light-dark transitions. Pharmacology Biochemistry and Behavior, 145, 55-65.

  • The authors may want to subdivide the discussion as to facilitate readership

The authors appreciated the constructive suggestion and strongly agreed on the necessity of subdividing the discussion section to facilitate readership. Therefore, a subtitle was given in each section of the discussion part to help the readers comprehend the manuscript.

Reviewer 2 Report

Comments and Suggestions for Authors

 General comment:

This is nice and rather well-designed study with potentially important repercussions on marine and freshwater ecotoxicology. The behavioral experiments include a large number of endpoints. These latters, when combined following the drug exposure, may characterize the phenotypic ethological damage.

The Authors may expand their conclusions, figuring which may be the actual ecological damage provoked by environmental exposure to this drug, in term of diminished fitness of the fish population.

Specific points:

Page 1

Lines 35-36

The results showed a relatively normal binding energy, indicating that this interaction might only partly contribute to the observed

“Binding energy”, especially in an Abstract, may result a rather obscure terminology.

Page 2

Lines 60-61

there are several adverse effects associated with its usage for long periods of time. Several adverse effects reported on the use of tacrolimus are

It is not immediately clear if the “several adverse effect” coincide or not with the following“Several adverse effects”

Lines 80-81

effects of tacrolimus in Siamese fighting fish which caused a significant behavioral impairment after tacrolimus treatment. Furthermore

Which is such a “behavioral effect” quoted for  Betta?

Line 89

innovative models to better understand the effects of tacrolimus in vertebrates [8]

Humans are of course Vertebrates, but for the biomedical community do represent a special case. I wonder if this may be rephrased.

Page 7

Lines 244-245

with ~2 liters of water inside an incubator to maintain the temperature at 26 ± 1°C and water pump to keep

24-26 C is a more usual range of temperature for Danio. However, they are naturally quite plastic.

Page 20

Lines 561-563In zebrafish, locomotion is a complex behavior produced by the activity of various neurons in the neurotransmitter systems, which are also common to that of other vertebrates

Actually, brain areas and\or  specific  circuits are more often quoted as the basis of locomotor patterns in vertebrates.

Page 21

Lines 626-629On the contrary, some behavior results of the lower dose of the tacrolimus-treated group were rather indicate potential anxiogenic effects of this compound on adult zebrafish. In the novel tank test, this treated group displayed similar exploratory behaviors

How such a behavioural alteration may affect survival\fitness? Which may be the changes at a populational level of exposed fish?

Page 23

Lines 724-727

In conclusion, the current study demonstrated the toxicities of tacrolimus in relatively low concentrations to zebrafish, especially in their behaviors and cognitive performances, in both larval and adult stages after acute and chronic exposures, respectively, with some alterations were occurred in a dose-dependent manner (Figure 12).

The reported changes in social cohesion, not usually enlisted among “cognitive” endpoints, should be mentioned.

Author Response

Comments and Suggestions for Authors

General comment:

This is nice and rather well-designed study with potentially important repercussions on marine and freshwater ecotoxicology. The behavioral experiments include a large number of endpoints. These latters, when combined following the drug exposure, may characterize the phenotypic ethological damage. The Authors may expand their conclusions, figuring which may be the actual ecological damage provoked by environmental exposure to this drug, in term of diminished fitness of the fish population.

Specific points:

Page 1 Lines 35-36. The results showed a relatively normal binding energy, indicating that this interaction might only partly contribute to the observed

“Binding energy”, especially in an Abstract, may result a rather obscure terminology.

Thank you for the suggestion. It is true that for some readers, the “binding energy’ term might be an obscure terminology, especially if it was put in the abstract. Therefore, a brief definition regarding the meaning of this term was added to the abstract, and the term was changed to “binding affinity” to provide a clearer term although the meaning of both terms are same.

Page 2 Lines 60-61. there are several adverse effects associated with its usage for long periods of time. Several adverse effects reported on the use of tacrolimus are

It is not immediately clear if the “several adverse effect” coincide or not with the following“Several adverse effects”

The authors appreciated the comment and agreed that the sentences were not well-written and thus, could cause confusion to the readers. Actually, the first “several adverse effects” referred to the toxicities of the chronic exposure of tacrolimus while the latter referred to the both acute and chronic adverse effects of tacrolimus demonstrated in the previous studies. Therefore, to avoid confusion, the sentences were rewritten.

Lines 80-81. effects of tacrolimus in Siamese fighting fish which caused a significant behavioral impairment after tacrolimus treatment. Furthermore

Which is such a “behavioral effect” quoted for Betta?

Thank you for the question. Actually, the “behavioral effect” mentioned in the text refers to a significant reduction in the fish aggression level, which was indicated by lower opercular display in a mirror test after tacrolimus treatment. According to the reviewer’s comment, this information was added to the manuscript to avoid confusion.

Line 89. innovative models to better understand the effects of tacrolimus in vertebrates [8] 

Humans are of course Vertebrates, but for the biomedical community do represent a special case. I wonder if this may be rephrased.

The authors thanked the reviewer for the comment. In this sentence, the authors aimed to highlight the potencies of the fish as an animal model to study the adverse effects of tacrolimus in better understating its effects on higher life-form vertebrates, including humans considering the advantages that are possessed by zebrafish as mentioned in the manuscript. However, as the reviewer suggested, the sentence was rephrased to provide clearer ideas regarding the usage of zebrafish in this case.

Page 7 Lines 244-245. with ~2 liters of water inside an incubator to maintain the temperature at 26 ± 1°C and water pump to keep 

24-26 C is a more usual range of temperature for Danio. However, they are naturally quite plastic.

Thank you for the comment. Actually, zebrafish are able to live in a high range of temperatures as the reviewer stated. Therefore, based on some literature research, the temperature range mentioned by the reviewer and mentioned in the manuscript is still optimal for zebrafish. However, the authors decided to select the current temperature based on the previous study mentioned below.

Avdesh, A., Chen, M., Martin-Iverson, M. T., Mondal, A., Ong, D., Rainey-Smith, S., ... & Martins, R. N. (2012). Regular care and maintenance of a zebrafish (Danio rerio) laboratory: an introduction. JoVE (Journal of Visualized Experiments), (69), e4196.

Page 20 Lines 561-563. In zebrafish, locomotion is a complex behavior produced by the activity of various neurons in the neurotransmitter systems, which are also common to that of other vertebrates

Actually, brain areas and\or  specific  circuits are more often quoted as the basis of locomotor patterns in vertebrates.

The authors appreciated the suggestion. It is true that the complex behaviors in vertebrates are regulated by neurons in the brain areas as the reviewer stated. Therefore, the sentence was revised according to the reviewer’s comment, highlighting the brain as the main basis of locomotor patterns regulator in vertebrates.

Page 21 Lines 626-629. On the contrary, some behavior results of the lower dose of the tacrolimus-treated group were rather indicate potential anxiogenic effects of this compound on adult zebrafish. In the novel tank test, this treated group displayed similar exploratory behaviors

How such a behavioural alteration may affect survival\fitness? Which may be the changes at a populational level of exposed fish?

Thank you for the questions. Actually, alterations in exploratory behaviors might affect the survival or fitness of fish since a well-developed exploratory behavior may provide benefits during a life stage when animals shift into a new habitat or disperse over a larger area if the mortality risk in the present habitat is increasing and/or prey abundance decreases. This information was added to the manuscript to highlight the importance of the exploratory behaviors in fish as the reviewer suggested.

Page 23 Lines 724-727. In conclusion, the current study demonstrated the toxicities of tacrolimus in relatively low concentrations to zebrafish, especially in their behaviors and cognitive performances, in both larval and adult stages after acute and chronic exposures, respectively, with some alterations were occurred in a dose-dependent manner (Figure 12).

The reported changes in social cohesion, not usually enlisted among “cognitive” endpoints, should be mentioned.

The authors thanked the reviewer for the comment. Initially, the authors thought that the alterations observed in the shoaling cohesion of the treated fish were represented in “behaviors” in the sentence. However, according to the reviewer’s suggestion, the authors decided to add “social cohesion” in the sentence to highlight the alterations caused by tacrolimus found in the present study.

Round 2

Reviewer 1 Report

Comments and Suggestions for Authors

Authors addressed  this reviewers's  concerns

Comments on the Quality of English Language

Would  revise  English  by native speaker  ( For example  lines 221-223) convey the meaning unusual  phrasing

Author Response

Would  revise  English  by native speaker  ( For example  lines 221-223) convey the meaning unusual  phrasing

We have done details grammar check for this 2nd revised article and all edited parts were hightlighted with Word checking system. We hope this revised version can reach the high standard of Biology journal.
